# The Economic Viability of a Progressive Smart Building System with Power Storage

**Eerika Janhunen \*, Niina Leskinen and Seppo Junnila**

Department of Built Environment, Aalto University, 14100 Aalto, Finland; niina.leskinen@aalto.fi (N.L.); seppo.junnila@aalto.fi (S.J.)

\* Correspondence: eerika.janhunen@aalto.fi

**Abstract:** The increased smartness of the built environment is expected to contribute positively to climate change mitigation through energy conservation, efficient renewable energy utilization, and greenhouse gas emission reduction. Accordingly, significant investments are required in smart technologies, which enable the distributed supply of renewables and increased demand-side energy flexibility. The present study set out to understand the cash flows and economic viability of a real-life smart system investment in a building. The data collection process was threefold: First, a case building's level of (energy) smartness was estimated. Second, the semi-structured interviews were held to understand the building owner's motives for a smart investment. Third, the investment's profitability was analyzed. The study found that the progressive smartness investment was technically feasible, and surprisingly also economically profitable. The original EUR 6 million investment provided over 10% return-on-investment and, thus, increased the property value by more than EUR 10 million. Moreover, the commercial partners also emphasized the strategic value gained by renewable energy and environmental performance. The high level of smartness with a good return on investment was accomplished mainly through new income generated from the reserve power markets. However, the results implied that financial profitability alone was not enough to justify the economic viability of a smart building system investment.

**Keywords:** smart building; smart energy system; renewable energy resources; energy storage; reserve power system; investor motives; investment profitability; smart readiness indicator; discounted cash flow analysis

## 1. Introduction

The world's population is estimated to increase by one-third in the next 30 years, to 9.7 billion in 2050. By then, an estimated 6.7 billion people will live in urban areas [1]. This predicted rapid urbanization could be considered as an opportunity, but it also presents a challenge to making cities resilient and sustainable in line with the United Nations' sustainable development goals [2]. Furthermore, such rapid socio-economic development will significantly affect the long-term outlook of energy, as the demand for space heating and cooling, for instance, will rise [3]. Therefore, it is vital to make buildings, both directly and indirectly, less energy- and carbon-intensive in the future [4].

The greatest challenges to achieving a decarbonized energy system and, indirectly, for the building stock are the efficient deployment of renewable energy sources (RES) and the use of the most efficient generation technologies [5,6]. The most promising solution for the sector appears to be the integration of the electricity network into buildings' energy systems [7,8]. The integration of information and communication technologies (ICT) in the energy system may be the key to achieving a decarbonized building stock and accelerating the energy system transformation [3]. The adoption of ICT will enable

a faster energy market operation that is more responsive to the balancing needs of a power system with less inertia and faster rates of change [9,10].

To support the energy system transformation, and to enhance the uptake of RES, the European Commission has strongly directed European Union (EU) members to engage in activities that promote the adoption of digital solutions in the built environment. One such activity is the development of a smart readiness indicator (SRI) [11]. The objective of the indicator is to provide an equal rating system for EU members and raise awareness of the benefits of grid flexibility enabled by distributed and fast-responding electricity and thermal storages, electric vehicles (EVs), and demand response. In alignment with the scope of the proposed framework, the SRI aims to evaluate a building's potential to optimize the overall energy consumption, provide occupants with more accurate information about their consumption, and enable the central system operators to manage the grid more effectively based on demand [12]. The SRI for buildings is not, naturally, an indicator of the maximum level of smartness in a building system. Nevertheless, it aims to provide a way to support the cross-sectorial integration of the building sector into (future) smart energy systems by enhancing the role of the building, the user, and the grid.

One of the key goals behind the development work of the SRI is to make the added value of building smartness more tangible for property owners. So far, however, evaluations of holistic smart energy investments, that support real estate investment valuations, are still lacking in the literature. Previous studies, such as [13–15], have mainly concentrated on measuring economic aspects of various stand-alone smart energy systems. In these studies, the financial profitability of the investment in a smart system has been estimated from a technology project perspective by using traditional economic analysis methods, including internal rate of return (IRR), return on investment (ROI), and payback period. However, even though the investments as such have appeared appealing, these frequently applied valuation methods do not consider the possible impact of such investments on property value.

From the real estate investment point-of-view, the property is evaluated as an entity with the focus on its total value [16]. The value of professionally managed investment properties is often evaluated using a discounted cash flow (DCF) analysis. In a DCF analysis, the present value of a property is based on the estimated cash flows and exit value, which are discounted to the present with a suitable discount rate [17]. The most important parameters forming the cash flow of a property are rental income, rental growth, vacancy rate, operating expenses, capital expenditure, depreciation, and a discount rate that reflects the relevant risks [17]. Depreciation includes both physical deterioration and obsolescence [18]. Thus, to understand the real estate investors' perspective and capture the value of smart building investments for them, DCF analysis should be applied in evaluating the economic profitability of such investments.

In the current literature, there is only a limited number of studies, if any, that consider the property value aspect of a smart (energy) system investment in a building. On the other hand, existing research that considers the property value aspect focuses purely on energy efficiency improvements (i.e., does not consider the system smartness). However, these studies mainly apply statistical analysis [19–21], which does not explain the value influencing mechanism of such investments in detail. Christersson et al. [16] and Leskinen et al. [22] seem to be the only practitioners who have considered the value-influencing mechanism of energy efficiency improvement investments of on-site energy production in a DCF framework. Additionally, Vimpari et al. [23,24] and Kontu et al. [25] considered the potential property value increase in their profitability analysis of rooftop solar and ground source heat pump investments. Interestingly, hardly any studies have estimated the financial feasibility of the technological shift towards smart energy systems at the property level.

The present study was designed as a novel case study that examines the economic viability and impact on the property value of a real-life smart building system investment. The implemented energy system generated not only traditional energy savings but also new income for the property through the participation in the frequency containment reserve (power) markets. This is the first study known

that has used empirical cash flow data and utilized property investment analysis to reveal the added property value of such a smart energy system in buildings.

The present study provides insight into a smart energy system investment in a case building through a technology description, investor interviews, and an investment's profitability analysis. First, the case building's smartness, i.e., its technological readiness to support the energy system transition, was assessed using the EU-driven SRI rating system. Second, the economic and strategic motives of the investment were identified through interviews with representatives of the case building's owner. Third, investment analysis with case-specific data was performed. To the best of the authors' knowledge, this is the first study to apply a property investment analysis to a real-life smart energy system investment.

The study found that the building system was highly advanced in terms of its energy smartness, signified by a near-maximum score on the smartness rating scheme (SRI). The core technologies for achieving a high score was the system's microgrid functionality, on-site energy capacity, and advanced demand management capabilities. In the interviews, representatives of the building's owner implied that the investment was justified mainly by decreased operating costs and income related to participating in the frequency containment reserve market, which improved the net cash flow of the property. However, the improved net cash flow and the lucrative internal rate of return (IRR) were not enough to make the investment appealing. Besides, the smart energy system supplier's, i.e., service supplier's, active, and service-oriented attitude appeared essential in investment decision-making. Finally, a government subsidy made the investment even more lucrative. The additional strategic value of "being smart and environmentally excellent" was also considered an important factor in executing the investment. The explicit reasoning of the more sophisticated drivers, such as branding and image benefits, were recognized, but their influence on investment's profitability was difficult to evaluate in financial terms.

The present paper is structured as follows: Section 2 describes the research design of the study, including the case description and empirical data collection methods. Section 3 reviews the empirical research results comprising the energy smartness assessment, semi-structured interviews, and investment's profitability analysis. Section 4 further discusses the results, and Section 5 presents the conclusion.

## 2. Research Design

The present study was designed as a descriptive case study to examine a real-life smart energy system investment that supports the cross-sectorial integration of the building sector into smart energy systems. This study aimed to investigate the value creation of investment from a real estate market perspective. In this section, the case building and data collection methods are introduced.

### 2.1. Case Building

The study case was chosen based on an extensive smart energy system investment implemented in the building in recent years. The system consisted of substantial energy storage, software development, and energy conservation technologies. The investment was funded by the consortium of three Finnish institutional investors. The case building, considered to be a prime investment property, was located in southern Finland, which is one of the few EU countries where smart technologies have been systematically implemented into the built environment [26]. In 2015, the case building was the first European shopping center to receive LEED Platinum certification for existing buildings. The key characteristics of the case building are shown in Table 1.

**Table 1.** Characteristics of the case building.

| Year of Constr. | Area [m$^2$] | Building Type | Year of Smart Energy Investment | Smart Technologies |
|---|---|---|---|---|
| 2003 | 100,000 | Commercial building | 2018 | PV system, battery storage, active LEDs, EV charging, advanced demand management (software development) |

The data collection process for the case study was threefold: First, the building system's level of (energy) smartness was measured using the EU-driven SRI rating scheme [12]. Second, semi-structured interviews were held with representatives of the case building's owner. Third, the investment's profitability was calculated using widely applied investment evaluation methods. In the present study, the profitability of the investment was additionally evaluated through the impact of the savings in operating expenses and additional income (generated by the investment) on the property value. In the following sections, the data collection methods are further reviewed.

*2.2. Energy Smartness Assessment*

The energy smartness of the case building was evaluated by using the EU-driven SRI rating scheme. Today, the SRI is still under development; therefore, the most recent scheme [12], which was available at the time of the assessment, was applied to measure the energy smartness of the building system. Here we will introduce briefly the applied assessment methodology of the rating scheme as well as describe the assessment setup. A more detailed description of the assessment methodology is provided by Janhunen et al. [27].

2.2.1. Assessment Methodology

The SRI rating scheme is based on the assessment of a predefined list of smart services, which are enabled by a set of smart (ready) technologies. The practical, i.e., streamlined, version of the service list is divided into 10 distinct main domains, which are the following:

1. Heating
2. Domestic hot water (DHW)
3. Cooling
4. Controlled ventilation (MV)
5. Lighting
6. Dynamic building envelope (DBE)
7. On-site renewable energy generation (EG)
8. Demand side management (DSM)
9. Electric vehicle charging
10. Monitoring and control (MC).

In the version, which was applied in the present study, these domains contain altogether 52 smart services, which are inspected as part of the assessment. Each service has been given various degrees of smartness, i.e., functionality levels, where the lowest functionality, level 0, refers to non-smart service implementation and the highest level refers to an adaptive functionality with a demand-based service control. Hence, the highest functionality level varies from service to service. Additionally, each listed service in the SRI scheme has a potential (positive or negative) impact on the building occupants, and/or the building itself, and/or the grid. These impacts have been categorized into eight categories: energy savings on-site, flexibility for the grid and storage, self-generation, comfort, convenience, wellbeing and health, maintenance and fault prediction, and information to available occupants.

The final SRI score, i.e., the level of energy smartness, is a result of a multi-criteria assessment, which leads to an explicit percentage expressing how close (or far) the building is from its theoretical maximum smartness. The maximum smartness is individual for each building. The multi-criteria assessment method, which was applied to calculate the case building's level of energy smartness, followed the methodology provided in the final report of the SRI's first technical support study [12].

2.2.2. SRI Case Assessment

The energy smartness assessment was conducted within the premises of the case building in March 2019. The assessment was performed in a workgroup consisting of the property manager,

the representative of the service supplier, and the SRI evaluation team members. The first author of this paper acted as the SRI assessment evaluator. In alignment with the SRI methodology [12], the assessment of those services, which were SRI compatible but not relevant, i.e., applicable, in the case building, did not affect the final scoring.

The assessment session began with a presentation of the SRI rating scheme. The assessment was performed using a qualitative checklist approach. The representative of the service supplier performed the role of a technical building systems (TBS) specialist in the assessment. The TBS specialist indicated the implemented functionality levels for the applicable smart ready services. The evaluation team inputted the scores into an excel-based calculation tool, which aggregated the overall SRI scores. The calculation tool was developed by the evaluation team following the applied SRI methodological framework [12]. Because the smart energy system was recently implemented and currently being operated by the service supplier, the TBS specialist was able to determine almost all of the functionality levels without consulting technical documents. Only a few service levels had to be checked on the documents, which were inputted into the calculation tool afterward. The workshop took approximately 1.5 h. The assessment did not include a walk-through inspection in the building's technical facilities.

### 2.3. Semi-Structured Interviews

Altogether, six semi-structured interviews were held with representatives of the case building's owner. The interviewees for the study were chosen based on their involvement in and knowledge of the investment decision-making and/or management phase of the smart energy system. The interviewee descriptions are visible in Table 2.

**Table 2.** Interviewee descriptions.

| Interview | Title of the Interviewee | Role in the Investment |
|:---:|:---:|:---:|
| A | Real Estate Portfolio Manager (former) | Primary owner, involved in the decision-making |
| B | Real Estate Investment Director (former) | Primary owner, involved in the decision-making |
| C | Real Estate Investment Manager | Primary owner, involved in the management phase |
| D | Sustainability Manager | Primary owner, involved in the management phase |
| E | Business Development Director | Owner, involved in the decision-making and management phase |
| F | CEO | Shopping center manager |

Four interviews were held with the primary owner of the case building. Two interviews (A and B) were held with the representatives involved in the investment decision-making phase, and two (C and D) with the representatives who had the best knowledge of the management phase of the smart energy system. To increase the validity of the interviews, one interview (E) was held with an owner representative who was involved both in the decision-making and management phases of the investment. The last interview (F) was held with a representative of the shopping center manager to obtain insights regarding the smart energy system's operational side.

The interviews were held by the first and second authors of this paper in February 2020, after the first full operational year of the smart energy system. Interview A was an exception: it was held in January 2018. Interview A provided evidence of the investment decision-making process before the smart energy system was fully operational and was applied as a preliminary research dataset for the present study. The interviews were conducted in Finnish.

The application of the interviews as a data collection method was twofold. First, the interviews were used to identify the key themes supporting the investment decision-making process. Second, the interviews were used to confirm the results of the conducted investment analysis calculations.

At the beginning of each interview, the SRI rating system was briefly introduced and the case building's assessment results were summarized. Thereafter, the interviewees were asked to describe the investment decisions and management phase of the smart energy system. Finally, the investment's

profitability calculations described below were shown and the interviewees were asked to comment on both the collected input data and the outcome of the calculations.

All the interviews were semi-structured and took approximately 1h each. The interview questions were not delivered to the interviewees beforehand. The interviews were recorded, and both interviewers made notes along with the discussion. Afterward, the notes were accumulated in a SharePoint environment, and the second author conducted a content analysis of the data set and categorized the results into themes. The first author validated the results by referring to the recordings and confirmed the substance of the key findings that arose from the analysis.

### 2.4. Investment's Profitability Analysis

The investment's profitability analysis was based on real investment data and new cash flows generated by the smart energy system. In this section, we introduce the applied investment data and describe the conducted analysis with a case-specific example.

### 2.4.1. Investment Data

The total investment cost amounted to approximately EUR 6 million. The investment was financially supported by a government subsidy of EUR 2 million. The implemented smart system consisted of the main technologies of the rooftop PV, energy storage, and system integration (including the software development of the advanced demand management capabilities). The investment generated both savings and new income.

The rooftop PV investment amounted to approximately EUR 600,000. Aligned with the recent researches, the life cycle of the PV system was assumed to be 30 years [28,29]. In economic analysis, the inverter replacement costs are often included in the operating expenses of a PV system, as explained by Vimpari and Junnila [23]. It is assumed that the life cycle of the battery was 20 years [6,30]; after that, the owner would invest in a new battery, the investment cost of which totals EUR 2 million [30].

Due to confidentiality reasons, real maintenance cost data was not available. In this paper, the maintenance costs were estimated from Finnish data [31] based on the relationship between technical property maintenance costs and investment costs. Based on this data, the maintenance costs of the system amounted to 1% of the investment costs. In addition to yearly maintenance costs, this estimate included insurance and repair costs.

The annual estimates of the savings and new income generated by the investment were based on 10 months of actually running the smart energy system. The savings were generated from the energy efficiency improvements and the new income from the reserve power markets enabled by the battery. The investment dataset, which was applied in the financial analysis, is summarized in Table 3.

**Table 3.** A summary of the investment data.

| Smart Energy System | | Investment [EUR] | Life Cycle [yr.] | Maintenance [EUR/yr.] | Savings [EUR/yr.] | New Income [EUR/yr.] |
|---|---|---|---|---|---|---|
| **Total** | | 6M | 30 | 60,000 | 180,000 | 480,000 |
| | Rooftop PV | 600,000 | 30 | N/A | 60,000 | N/A |
| | Battery | 2M | 20 | N/A- | N/A | 480,000 |
| | System integration | 3.4M | N/A | 60,000 | 120,000 | N/A |

Note: system integration means development, design, integration, and maintenance.

### 2.4.2. Description of the Analysis

The financial profitability of the smart energy system investment was first evaluated from a technological project perspective by applying the static investment metrics of payback period and ROI. However, as these metrics do not capture the time value of money nor the lifetime of the investment, the IRR was calculated using a spreadsheet program with a 30-year life cycle. A 30-year life cycle was

selected based on the typical life cycle of the installed technical elements. The payback period, ROI, and IRR were calculated with and without the government subsidy using the equations explained below.

The payback period was calculated using the following equation [32]:

$$Payback\ period = \frac{Investment\ costs}{Annual\ net\ cash\ flow} \tag{1}$$

where investment costs equal the total cost of the investment, and the annual net cash flow equals the yearly amount of income and savings in operating costs generated by the investment.

The ROI was calculated using the following equation [33]:

$$Return\ on\ investment = \frac{Profit}{Investment\ costs}, \tag{2}$$

where profit equals the income and net savings generated by the investment, and investment costs equal the total cost of the investment.

The IRR of the investment was calculated using a spreadsheet program that uses the following equation [34]:

$$Net\ present\ value\ (NPV) = \sum_{i=1}^{30} \frac{CF_t}{(1+d)^i} = 0, \tag{3}$$

where $CF_t$ denotes to cash flows (i.e. the net savings and income) in different years; and $d$ is the discount rate, which equals the IRR when NPV is zero.

Even though the payback period, ROI, and IRR are widely applied investment evaluation methods, they do not consider the positive impact of a smart energy system investment on property value. Thus, in the present study, we applied a DCF framework to support the conducted investment analysis. By applying the DCF framework, the profitability of the investment was evaluated based on the new cash flows (generated by the smart energy system) on property value. The cash flows consisted of the savings in operating expenses and additional income (associated with the battery). By using the DCF framework, the present value of a property can be expressed as follows with a 30-year life cycle [35]:

$$Present\ value\ of\ property = \sum_{i=1}^{30} \frac{(Gross\ income - operating\ expenses)}{(1 + property\ yield)^i} \tag{4}$$

The above equation clearly shows that a decrease in operating expenses leads to an increase in the value of the property through the capitalization of the improved net cash flow, as the International Valuation Standards suggest [35]. Leskinen et al. [36] described the impact of the value-influencing mechanism of on-site energy investment (which can be assimilated into smart building investments) on property values and justified the use of property yields as discount rates in these kinds of investments. Accordingly, the increase in property value generated through the savings in the operating expenses and additional income can be expressed as follows:

$$Property\ value\ increase = \sum_{i=1}^{30} \frac{CF_t}{(1 + property\ yield)^i} \tag{5}$$

Between 2000 and 2018, electricity prices increased faster than inflation in Finland. From the property owners' perspective, the faster increase in energy prices compared to the rental growth rate motivated investment in self-generated energy production to protect the property from the risk of rising energy prices. Between the period, the increase in electricity prices amounted to 4.1% p.a. [37], while the increase in consumer price index totaled 1.5% p.a. [38]. In the analysis of this paper, 1.5% was used as the inflation rate. As cash flows included the expected inflation, the net savings and income of the property value increase function were discounted with the sum of the area's prime retail yield

of 4.5% [39] and an inflation rate of 1.5% [38]. The electricity price growth rate, instead, was used as the inflation rate when estimating the savings generated by the PV system. The applied discount rate variables, which were applied to calculate the property value increase, are shown in Table 4.

**Table 4.** The applied discount rate variables for calculating the property value increase.

|  | PV Savings | Battery Income | System Int. Savings | Maintenance Costs | Total $CF_t$ |
|---|---|---|---|---|---|
| **Discount rate (d)** | 4.1% | 1.5% | 1.5% | 1.5% | 6% |

For the sake of clarity, we show the property value increase equation by using the investment data (Table 3) and the applied discount rate variables (Table 4) Although the investment consisted of different parts (PV system, battery, and system integration), the investment was evaluated as one entity as suggested by the interviewees. The expected property value increase generated by the smart energy system investment was calculated as follows:

$$
\begin{aligned}
\text{Property value increase} &= CF_0 + \sum_{i=1}^{30} \frac{CF_t}{(1+d(CF_t))^i} - \text{Battery capex}(i=20) \\
&= (\text{PV savings} + \text{battery income} + \text{system int. savings} - \text{maintenance costs}) \\
&\quad + \sum_{i=1}^{30} \frac{\text{PV savings} * (1+0.041)^i + (\text{battery income} + \text{system int.savings} - \text{maintenance costs}) * (1+0.015)^i}{(1+0.06)^i} \\
&\quad - \frac{\frac{\text{investment (battery)}}{(1+0.015)^{20}}}{(1+0.06)^{20}} \\
&= (60{,}000 + 480{,}000 + 120{,}000 - 60{,}000) \\
&\quad + \sum_{i=1}^{30} \frac{60{,}000 * 1.041^i + (480{,}000 + 120{,}000 - 60{,}000) * 1.015^i}{1.06^i} - \frac{\frac{2M}{1.015^{20}}}{1.06^{20}} \\
&= 600{,}000 + \sum_{i=1}^{30} \frac{60{,}000 * 1.041^i + 540{,}000 * 1.015^i}{(1.06)^i} - \frac{\frac{2M}{1.015^{20}}}{1.06^{20}}
\end{aligned}
\tag{6}
$$

## 3. Results

In this section, the results from the case study are introduced. First, the results from the SRI assessment are presented. Second, the key investment motives that arose from the semi-structured interviews are introduced. Third, the investment's profitability analysis results are shown.

### 3.1. Smartness Evaluation of the Building and Relevant Technologies

In this study, the SRI rating system was applied to identify the energy smartness of the building. In this section, the results from the energy smartness assessment are reviewed.

The case building's final score was 92% of the maximum on the SRI rating scale, which indicated that the building was indeed exceptionally smart in terms of its technological implementations. From 10 domains listed in the SRI framework, nine were identified as present in the case building. Only the main domain of DBE was not implemented in the building. In total, 39 smart (ready) services from the list of 52 were identified as applicable in the assessment. The applicable services, their levels of energy smartness (%), and brief descriptions of primary technologies are shown in Table 5.

**Table 5.** The energy smartness of the applicable smart services.

| Service | Smart Technology | Score | Service | Smart Technology | Score |
|---|---|---|---|---|---|
| Heating-1a | Individual room control with communication and presence control | 100% | MV-2d | Variable set point with load-dependent compensation | 100% |
| Heating-1c | Demand based control | 100% | MV-3 | Free cooling | 67% |
| Heating-1d | Variable speed pump control (external demand signal) | 100% | MV-6 | Real-time information of indoor air quality available to occupants and suggesting triggers to action | 100% |
| Heating-1e | Automatic control with demand evaluation | 100% | Lighting-1a | Automatic detection (manual on/dimmed or auto-off) | 100% |
| Heating-1g | Program heating schedule in advance | 50% | Lighting-2 | Scene-based light control | 100% |
| Heating-2a | Variable temperature control depending on outdoor temperature | 50% | EG-2 | Performance evaluation including forecasting and/or benchmarking; also including predictive management and fault detection | 100% |
| Heating-2c | Load prediction-based sequencing | 100% | EG-3 | Dynamically operated storage which can also feedback into the grid | 100% |
| Heating-3 | Performance evaluation including forecasting and/or benchmarking; also including predictive management and fault detection | 100% | EG-4 | Long term optimization including predicted generation and/or demand | 100% |
| DHW-3 | Performance evaluation including forecasting and/or benchmarking; also including predictive management and fault detection | 100% | DSM-18 | Building energy systems are managed and operated depending on grid load; demand side management is used for load shifting | 100% |
| Cooling-1a | Individual room control with communication between controllers and to building automated control system | 75% | DSM-19 | Smart appliances, DHW, heating, and cooling subject to DSM control | 100% |
| Cooling-1c | Demand based control | 100% | DSM-21 | Reporting information on current, historical and predicted DSM flows and controls | 100% |
| Cooling-1d | Variable speed pump control (external demand signal) | 100% | DSM-22 | Scheduled override of DSM control and reactivation with artificial intelligence | 100% |
| Cooling-1e | Automatic control with demand evaluation | 100% | EV-2 | Medium charging capacity | 67% |
| Cooling-1f | Total interlock | 100% | EV-16 | One-way (controlled charging) | 50% |
| Cooling-2a | Variable temperature control depending on the load | 100% | EV-17 | Communication with a back-office compliant to ISO 15118 | 100% |
| Cooling-3 | Performance evaluation including forecasting and/or benchmarking; also including predictive management and fault detection | 100% | MC-3 | Individual setting following a predefined schedule; adaptation from a central room; variable preconditioning phases | 67% |
| MV-1a | Demand control based on air quality sensors | 100% | MC-4 | With central indication of detected faults and alarms/diagnosing functions | 100% |
| MV-1b | Variable control | 100% | MC-9 | Occupancy detection for individual functions, e.g., lighting | 50% |
| MV-1c | Automatic flow or pressure control (without reset) | 75% | MC-13 | Real-time indication of sub-metered energy use or other performance metrics for all main TBS | 100% |
| MV-2c | Modulate or bypass heat recovery based on multiple room temperature sensors or predictive control | 100% | | | |

The case building scored 100% in 30 (out of 39) smart ready services on the SRI rating scheme. It scored less in nine services, scoring between 50% and 75%. The average SRI score was 91% (Table 5). In the service categories in which the building scored less than maximum, the upgrade to the maximum would have required the implementation of the following smart technologies: thermostat self-learning user behavior (Heating-1g), load-based control (Heating-2a), presence control (Cooling-1a), control

with reset (MV-1c), H,x-directed control (MV-3), high charging capacity (EV-2), two-way balancing (EV-16), control of run-time management by artificial intelligence (MC-3), and centralized detection feeding into several TBS, such as lighting and heating (MC-9). The more sophisticated explanations of the listed SRI compatible smart services and their related technologies can be found in the final report of the first technical support study [12].

### 3.2. Investor Motives Regarding the Smart Building System Investment

The semi-structured interviews elucidated on the decision-making process of the smart energy system investment. The investor motives were assigned to key categories as identified in the content analysis of the interviews. The analysis delivered the following key themes, along with the improved net cash flow of the property and appealing IRR: image benefits, mitigation of the environmental and energy price risks, and solid trust in the long-lasting collaboration with the service supplier. The investment was also found to have some risks, which are shortly described in this section.

### 3.2.1. Enhanced Image

A majority of the interviewees mentioned that the strategy of the shopping center was to be a forerunner in environmental issues. Both of the interviewed investor representatives confirmed that they had signed a responsible investment commitment at the company level. However, the enhanced image was not only seen as an important means of engaging visitors to the shopping center, but also as a way to attract new tenants and improve the likelihood of renewing leases with the current tenants.

### 3.2.2. Future Price Risk Mitigation

All the interviewees mentioned that the investment could also be seen as cutting long-term maintenance costs and following liabilities to repair the property. Mitigating environmental and energy price risks was also seen as an important reason for the investment. Protecting the property from the risk of rising electricity prices was mentioned by some of the interviewees. In addition to electricity price growth risk, some of the interviewees mentioned that enhancing sustainability and energy self-sufficiency protects the cash flow and exit value of the case property from tightening environmental regulation. For instance, a possible future carbon tax might apply directly to properties and increase maintenance costs. The investment was seen as a means of protecting the property rising electricity prices and the financial consequences of possible environmental regulation. Two of the interviewees mentioned that the electricity price growth risk and possibly changing energy fee structures and taxes might affect the estimated profitability of the investment. However, they felt that it was more likely that environmental regulations would tighten, causing taxes and energy fees to rise, which would improve the profitability of the investment.

### 3.2.3. Long-Lasting Collaboration with the Service Supplier

All the interviewees mentioned that the investment was originally introduced to the owners by the service supplier, whose active role was one of the most important factors driving the owners to execute the investment. Furthermore, the supplier actively aimed to increase the owners' confidence in the investment by committing to the project in the form of a long-term service agreement and sharing the (economic) risks with the owner. The active role of the shopping center manager was also mentioned during the interviews. The interviewees noted that the previous nearly 10-year working relationship with the service supplier (related to energy management of the case property), the supplier's credible track record, and resources were important parameters in the investment decision-making process.

### 3.2.4. Investment Risks

The interviewees were also asked to analyze the most significant obstacles and the most relevant risks related to the investment. All the interviewees mentioned the risk of new technology and risks

related to the estimated savings and profitability of the investment. Income-related to participating in the frequency containment reserve market (later referred to as battery income) was seen as a central source of uncertainty. One of the interviewees mentioned that one major risk (measured by its consequences) was whether the national main grid operator would allow the shopping center to participate in the frequency containment reserve market. All the interviewees mentioned that there was great uncertainty related to the (yearly) amount of battery income. Furthermore, the reputational risk was mentioned. Interviewees saw that reputational risks mainly consisted of the consequences of possibly realizing technical risks. Juridical risks related to the service agreement with the smart system supplier were carefully considered before the final investment decision was made. Although the investment was relatively small compared with the total annual maintenance costs of the property, consultants with expertise in the fields were asked for a second opinion regarding the technical and juridical risks.

### 3.3. Financial Profitability of the Smart Building System Investment

From the investor interviews, it was found that investment's profitability was the most important decision-making rationale. Hence, an analysis was done to justify the profitability of the investment. The calculation results were validated as part of the semi-structured interviews, where the interviewees were first asked to describe their investment analysis and then to comment on the analysis performed by the authors. Due to uncertainty related to the electricity price growth rates, battery income, and the maintenance costs of the system, a sensitivity analysis was performed for the conducted property value increase evaluation. This section introduces and describes the results of the financial investment analysis.

#### 3.3.1. Base Scenario

The investment's profitability was analyzed using three widely used investment evaluation methods in real estate economics: payback period, ROI, and IRR. Based on the economic analysis, two scenarios for the investment's profitability were formulated. The first scenario was calculated without the EUR 2 million government subsidy, and the second one with the subsidy. The resulted investment's profitability metrics of both scenarios are shown in Table 6.

**Table 6.** Investment's profitability metrics.

| Without the Subsidy | | | With the Subsidy | | |
|---|---|---|---|---|---|
| IRR | ROI | Payback Time | IRR | ROI | Payback Time |
| 5% | 10% | 10yr. | 11% | 15% | 6.7yr. |

Two of the interviewees mentioned that the payback period of the investment was under 10 years, which is in line with our results. Due to confidentiality reasons, the interviewees could not comment on the exact profitability metrics that were estimated in the investment decision phase. However, two of the interviewees mentioned that the IRR was appealing compared to the return of the property. This would be fulfilled in both of the scenarios, as the retail property yield (return) in the area was 4.5%.

Additionally, the investment's profitability was evaluated based on the capitalized value of the savings in operating expenses and the income related to the system by applying the DCF framework. Based on the analysis, the property value would increase by EUR 10.2 million. This means that the owner of the case property would immediately gain a benefit of over EUR 4 million from the investment. Next, we further reflect on some uncertainties that were found to concern the conducted property value increase analysis.

### 3.3.2. Sensitivity Analysis

To increase the validity of the primary financial analysis, i.e., the base scenario, a sensitivity analysis was performed for the expected property value increase. The sensitivity analysis concerned the savings generated by the PV system, the new income generated by the battery, and the maintenance costs.

First, an analysis of different electricity price scenarios was conducted to address the uncertainty related to the increase in electricity prices. The electricity price scenario was relevant to the savings generated by the rooftop PV system as it was the only element that was tied to energy price increase (and not inflation). In the base scenario, electricity prices were assumed to grow at the same rate as they did between 2000 and 2018. In the high electricity price scenario prices were assumed to grow 2%-point faster than in the base scenario.

Another important source of uncertainty was caused by the income associated with energy storage. In the base scenario, it was assumed that the battery would generate the estimated income for the whole life cycle of the investment. In the low-income scenario, it was assumed that only 80% of the estimated income would be received. In the high-income scenario, it was assumed that the income was 20% higher than estimated.

Finally, as the maintenance costs were estimated based on Finnish data [31], the appraise presented noteworthy uncertainty. In the low-costs scenario, the estimated maintenance was 80% of the estimated costs. In the high-cost scenario, the estimated maintenance was twice as much as estimated. The expected property value increase scenarios are summarized in Table 7.

**Table 7.** The results from the sensitivity analysis.

| Scenario | Property Value Increase [MEUR] | | | IRR (with the Subsidy) | | |
|---|---|---|---|---|---|---|
| | PV Savings | Battery Income | Maintenance Costs | PV Savings | Battery Income | Maintenance Costs |
| Base | 10.2 | 10.2 | 10.2 | 11% | 11% | 11% |
| Low | 9.8 | 8.5 | 10.4 | 11% | 8% | 11% |
| High | 10.6 | 11.8 | 9.2 | 11% | 13% | 9% |

## 4. Discussion

This study was designed as a descriptive case study to examine the viability of a progressive smart building solution that supports the cross-sectorial integration of the building sector into smart energy systems. The solution, including energy storage, software development, and energy conservation technologies, was considered as a real-life smart energy system. The viability of the investment was observed from the real estate market perspective, as evaluations of holistic investments in smart building solutions are still lacking in the literature. The study aimed to show the technological implementations, investor motives, and investment profitability of a smart energy system, using a market-driven case example with real-life data. The study found that investment in progressive smart building systems is already an economically viable option for contributing to the transition towards future smart and renewable energy systems. However, it was also found that the investment's profitability alone was not enough to justify such an investment.

In this study, the case building system's level of energy smartness was verified using the EU-driven SRI framework. Based on the energy assessment results, the case building was considered to be a real-life example of a viable smart energy system. The building's final score, over 90% of the maximum on the SRI scale, implied that sophisticated TBS appliances and technologies—which positively affect the building, the occupant, and the grid—have been implemented in the building [12]. Other smart technologies, including the PV system, active LED lighting, and EV charging were found to support the high SRI score significantly. Nevertheless, the power storage with the smart building's advanced demand management capabilities was considered to be at the core of the high scoring, as the relevance of grid flexibility has been strongly emphasized in the SRI development work [27,40]. Namely, the SRI rating scheme appears to increasingly favor demand response related features in buildings [40]. As has been shown, an integrated demand management system enables the efficient utilization of available

resources within the building system; it also integrates the building into the national energy system by acting as a reserve power system for the grid [41–43].

In this study, the economic viability of the investment was analyzed both from the qualitative and quantitative perspectives. The investor interviews revealed that the financial profitability of the investment was the most important rational in decision-making, but surprisingly not enough to justify the investment. As the investment's profitability analysis results implied, the owner of the case property would immediately gain a benefit of over EUR 4 million from the investment. Accordingly, compared to traditional investment evaluation metrics, the investment seemed to be highly appealing. Based on the interviews, the conducted analysis was seen as relevant and interesting. However, surprisingly this kind of property value increase analysis, which was performed in this paper, was not performed in the owner's investment decision-making phase.

In alignment with the EU's vision for future energy systems, buildings will have a crucial role as active energy prosumers in the transition to a decarbonized energy system by 2050 [44]. Hence, to support the transition and the efficient deployment of distributed RES, it will be critical that buildings all over the world be built according to the highest energy efficiency standards [3,45]. Based on the results of the present study, one of the key obstacles to the transition, however, appears to be the property investors averse to take a risk in smart investments.

Overall, the new technology-related risk is generally known to be one of the barriers of smartness in buildings [46,47]. In the present study, the motives to implement the smart technologies and advanced demand management system were found to be rather energy- and sustainability-driven, and the smartness itself was not considered as a value driver. Despite the improved net cash flow and lucrative IRR (compared to the area's retail property yield) generated by the investment, the most crucial part in the investment decision-making appeared to be the service supplier's active role and commitment to the management and development of the system, as well as their willingness to share part of the risk.

In the present study, the value increase of the case property was analyzed from the perspective of decreased operating expenses and new income generated by the battery. The value-influencing mechanism of a similar investment that enhances sustainability and decreases the operating expenses of properties was confirmed by surveyors in a study by Leskinen et al. [22]. In addition to the capitalization of operating expense savings, the value of a property can increase through other improvements in a property's cash flow parameters. Based on earlier research, property owners can benefit from investments that enhance the sustainability of properties through increased rent levels, rental growth, and occupancy rates, as well as decreased risks [48]. These enhancements can increase the property value even more than the capitalization of operating expense savings. However, earlier studies found that surveyors did not fully transfer these benefits into property values [22,49,50].

Although in the present study the smart energy system investment appears to be very appealing from a property value perspective, investors might not be able to execute the investment based solely on the estimated increase in the value. First, the value increase is hypothetical unless the investor sells the property, or an objective surveyor confirms the value. In practice, surveyors might not be able to reflect the decrease in operating expenses fully in the value of the property. They might need actual data on the decreased operating expenses for several years to verify the justified amount of savings. Irrespective of the investment, other cash-flow parameters might change, which could diminish the value increase resulting from the decreased operating expenses.

Secondly, investors traditionally focus on managing the income side of cash flow rather than optimizing operating costs. The share of operating costs amounts to approximately 5–15% of total cash flows [16], of which energy costs represent some 30% [51]. Although energy costs are a significant factor in the operating expenses of a property and have huge savings potential, they represent only a small share of the overall cash flow. Hence, the value increase potential is rather small compared to the overall value forming of the property.

Third, investing in and maintaining smart technology systems require special expertise that property investors might not have. Therefore, even though in this study the reserve power system was found to generate a significant potential for value increase, the uncertainty related to the income and new technology-related risk negatively affected the investors' expectations and willingness to invest in smartness. This should, however, create new business opportunities for technology service providers, as their relevance in maintaining and developing smart building systems can be expected to increase in the future.

In the case property, lease agreements follow a net lease structure, which means that tenants pay rent for maintenance on top of capital rent. Due to the savings in operating expenses, tenants might be able to pay higher capital rent, as it is the total amount of rent that matters from the tenants' perspective. However, the length of time needed for the rent levels to rise in practice is unclear. In a gross lease structure, where the owner of a property is responsible for operating expenses, the owner will immediately gain the benefits of the savings in operating expenses. In the end, the owner of the property will extract the same value from the property if the difference between net and gross rents equals the difference in operating expenses [52].

To the best of the authors' knowledge, this paper was the first study to apply a property investment analysis to a real-life smart energy system investment in a building. Hence, some limitations and uncertainties related to the results of the case study were identified. The greatest uncertainties and limitations were found to be related to the financial profitability of the smart energy investment. The annual savings were based on 10 months of actually running the system and on estimates provided by the service supplier. Actual savings can be very different from the estimate and can vary year to year. The most uncertain part of the savings is the income associated with the battery. This uncertainty was reflected in the sensitivity analysis, which contained three scenarios for the battery income. Furthermore, the electricity growth rate and inflation utilized in this study were based on the historical yearly average between 2000 and 2018. These growth rates can change over time.

Accordingly, a sensitivity analysis was added to show how the work could be improved and how a more realistic picture could be captured from the profitability analysis of the investment. In addition to growth rates, electricity prices and taxes can change over time, which might affect profitability. Besides, there were likely service charges paid by the owner to the service supplier that was not available for this study. These service charges might decrease the profitability of the investment. However, these kinds of charges might also include all the fees related to the maintenance of the technical system; therefore, this might have a minor or no impact on the results of the case study. Furthermore, this study did not consider possible enhancements in other cash flow parameters (a potential increase in rent, rental growth, and occupancy, as well as a decrease in discount rate) that could increase the value of the property even more than the decreased operating expenses.

Some limitations were found to concern also the other data collection methods. First, the SRI rating system applied to evaluate the case building's smartness is still under development; thus, the predefined list of smart services, as well as the functionality levels and impact weightings, are expected to change in the final version of the rating scheme. Additionally, the subjective decision-making related to the selection of an applicable service may affect the reliability of the assessment, as it has been explained by Janhunen et al. [27]. Secondly, some limitations were linked to semi-structured interviews. A majority of the interviewed investors were representatives of the primary owner, which might bias the results. However, the selected interviewees were considered to have the best understanding of the investment. To increase the validity of the results, one interview was also held with an independent secondary owner representative, who was involved both in the decision-making and management phase of the investment.

## 5. Conclusions

This study examined the economic viability of a real-life smart energy system investment in a building. The implemented system, including energy storage, advanced demand side management

(i.e., software development), and energy conservation technologies, was considered as an exemplary smart system solution that supports the future energy system transformation. The results of this study revealed that buildings' have the economic capability of becoming extremely smart to promote the cross-sectoral integration of the building sector into (future) energy systems.

From a real estate market perspective, there are multiple reasons to invest in smart technologies, including energy efficiency and lower operating costs with a predictable decrease in maintenance costs. However, the current study was the first in the smart building literature to evaluate the potential impact of smartness on property value through savings in operating expenses and additional income, specifically in the context of energy storage systems and new cash flows from the reserve power markets. The study found that even a progressive smart building system investment was economically profitable, and the investment generated over 10% return-on-investment along with over EUR 10 million increase in property value. However, the investment decision-making in smartness was not justifiable solely based on the appealing investment metrics, as the new cash flow opportunities were found to contain investment risks and practical challenges. For example, it is still uncertain how the property valuators approach the expected increase in the property value of such an investment.

Overall, due to the symbiotic nature of smart energy systems, the present study suggests that investment cash flows on a property level should be evaluated as one entity, instead of being broken down into subsystems based on smart technologies. Furthermore, the profitability of smart building investments should be evaluated through the impact of the savings in operating expenses and additional income (generated by the investment) on the property value to reveal the added value of smartness for property owners. However, further studies that consider the financial gains of the total smart energy system should be conducted to enhance the viability of the proposed solution as an option towards a renewable and sustainable energy system.

**Author Contributions:** Conceptualization, E.J. and S.J.; methodology, E.J. and N.L.; software, E.J. and N.L.; validation, N.L. and S.J.; formal analysis, E.J. and N.L.; investigation, E.J. and N.L.; resources, E.J.; data curation, E.J.; writing—original draft preparation, E.J.; writing-review and editing, E.J., N.L. and S.J.; visualization, E.J.; supervision, S.J.; project administration, E.J.; funding acquisition, S.J. All authors have read and agreed to the published version of the manuscript.

**Funding:** This research was supported by Aalto University's Smart Readiness Indicator project funded by the Finnish Ministry of the Environment, the Electrical Contractors' Association of Finland, the Finnish Electrotechnical Trade Association and the Finnish Association for Electrical Safety, and the Smart Land project funded by the Academy of Finland [13327800].

**Acknowledgments:** The corresponding author received personal funding by a Finsif scholarship.

**Conflicts of Interest:** The authors declare no conflict of interest.

## Abbreviations

The following abbreviations were used in the paper:

| | |
|---|---|
| DBE | dynamic building envelope |
| DCF | discounted cash flow |
| DHW | domestic hot water |
| DSM | demand side management |
| EG | renewable energy generation |
| EV | electric vehicle |
| ICT | information and communication technologies |
| IRR | internal rate of return |
| MC | monitoring and control |
| MV | controlled ventilation |
| NPV | net present value |
| PV | photovoltaics |
| RES | renewable energy sources |
| ROI | return on investment |
| SRI | smart readiness indicator |
| TBS | technical building systems |

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
