# Peer review of "The Economic Viability of a Progressive Smart Building System with Power Storage"

_sustainability, doi:10.3390/su12155998_

Round 1

Reviewer 1 Report

The mathematical functions used in section 2.4.1. Investment metrics need to be clearer in the application, I suggest use cases with each function or describe it better.

Conclusions, I suggest the authors describe in more detail what could be improved in the research. Evidence with more accurate data on the results obtained, justify the statements made in the summary.

References review
example missing link and/or reference location 1
Other online references, update access data.

Author Response

Thank you for your comments.

  • We have now aimed to clarify the application of the investment metrics and refined our main findings along with future research proposals.

2.4. Investment’s profitability analysis (In research design -section)

  • In Section 2.4.2. we have now further elaborated on the conducted investment analysis. Additionally, we provided a practical example of the property value increase function with case-specific variables on lines 302-306.

5. Conclusions

  • We have now added a more accurate description of the results. Additionally, the two last paragraphs (starting from line 564) provide several ideas how the research could be improved.

References

  • Thank you for the notice; we checked the required references 
  • We also updated the access data for online references

Reviewer 2 Report

In this paper, the authors successfully presented a technico-economic study of a virtual power plant implemented on a smart-building system. Through the presented case study, investment profitability and economic viability are analyzed and discussed. This study is very important to the literature since it includes not only the impact of the savings in operating expenses but also the additional income generated by the investment on the property value.

Author Response

Thank you for encouraging comments. 

Reviewer 3 Report

The paper deals with a fairly interesting topic.

The structure and detail requires significant improvement, though.
It is not clear, how the SRI score was derived - not from reference [28] either.
It just says that the SRI score is a result of multi-criteria assessment with no explicit calculation.

In Section 2.4, it is not clear how the results were derived from the equations presented before.

The findings of the interviews are presented as continuous text, which makes it hard for the reader to get a clear picture.
Tables or diagrams summarising the findings would have been helpful.

The results section includes a lot lof discussion and additional information.
This should be moved to another section.

I  recommend to prepare new paper with clear derivations and strict separation of scientific work and rather "subjective" information (e.g. findings from interviews), and to define a clear structure that makes it easier for the reader to follow.

Author Response

Thank you for your comments.

  • We have now aimed to clarify the overall content of the paper as well as refine the presentation of the results.

2. Research design  

  • Section 2.2.: We added the reference, which explicitly elaborates on the multi-criteria assessment method in detail. The methodology of the method is described step by step in the final report of the SRI’s 1st technical support study, which is referred on line 164.
  • Section 2.4.: We elaborated further on the conducted investment analysis. Additionally, we added a practical application of the property value increase function with case-specific variables on line 302-306.

3. Results

  • We divided the section 3.2. into sub-chapters to clarify the content of the results. 
  • We aimed to refine the results-section, especially the section 3.3., to focus on key findings and results. The more speculative content was moved to the discussion.

Summary

  • We have now aimed to make a clearer distinction between the results and discussion to clarify the content.
  • However, we found the qualitative part of the paper as crucial in terms of the research to validate the conducted investment analysis.
  • Accordingly, we changed the name of the paper to better accommodate the empirical (quantitative + qualitative) nature of the work.

Reviewer 4 Report

The article proposes an interesting study to check the viability of building intelligent energy environments that are respectful of the environment. The authors explain very well the work done. However, they only apply different existing analyses to determine whether a building is viable as an energy-efficient building. In other words, the research does not generate new content to science, or at least its contribution is not clearly reflected in the article. On the other hand, it is necessary to talk about certain aspects that need to be incorporated, clarified, or improved. These are described below.

- It is necessary to include a section where similar works are mentioned, analyzing the differences with the current study or including aspects that the proposal improves with respect to those studies. In this way, it will be possible to know how the proposal improves on those that already exist.

- Line 116 Do not include who is funding the research. Include this information in the funding section at the end of the article.

- In section 2.3 it would be very interesting to show the questions asked to the interviewees.

- Line 303 Further analysis is required of the new technologies to be implemented such as cost, implications it may have on the building, on the users, on the environment, etc.

Author Response

Thank you for your valuable comments.

  • The aim and originality have been now restated (and the title of the paper refined) so that it does not give a misleading impression of being a virtual power plant study, but a novel economic investment analysis of a real-life smart building with identified new cash flows.
  • We have now aimed to make a clearer distinction between the results and discussion to clarify the content. Additionally, we added a more accurate description of the results in the conclusion section. 

Section 1: Introction

  • The content of the paper has been refined to concentrate on the economic analysis of the smart investment.
  • The popular methods to solve similar problems have been reviewed, including payback period and return on investment (on lines 63, 65 and 66).

  • However, to the best of our knowledge the discounted cash flow analysis (DCF) has not been applied in similar (smart energy building) studies before

  • The benefits of the applied method (DCF) have now been better discussed (e.g. on lines 58-72) and it has now been emphasized how the work is the first one to apply the method in the smart building literature, specifically in the context of energy storage and cash flows from reserve power markets.

Line 116

  • Line 115 (in the revised manuscript) only states who invested in the smart energy system solution into the case building (Finnish institutional investors). The research (made by the authors) has been funded by third partys (Academy of Finland and a project supported by Aalto University) with no linkage to the investment studied in the present paper.
  • The funding details received by the present study have been explicitly introduced on lines 582-585. 

Section 2.3.

  • Thank you for your suggestion.
  • However, due to the nature of the interviews, the exact same questions were not asked from each interviewee. Some of the interviewees had a better knowledge of the investment-decision making and some had more knowledge from the operational side of the implemented system. The main content of the interviews has been described (starting from line 205).

Line 303, line 323 (in the revised manuscript) 

  • The content (and title) of the present paper has been refined, and the techno-perspective was left out of the scope to make the content more explicit.
  • In the refined content, we applied the SRI assessment only to validate the (energy) smartness of the building system. Therefore, we did not further analyze the content of the SRI, nor the technical implementations made in the case building.

Reviewer 5 Report

I would like to see the study of virtual power plants in the smart building system pay more attention to the problems of ensuring the safety, efficiency and reliability of the objects being created, and not just the amount of investment.

Author Response

Thank you for your comments.

  • The aim and originality have been now restated so that it does not give a misleading impression of being a virtual power plant study, but novel economic investment analysis of a real-life smart building with identified new cash flows.

  • In the present paper, it has now been emphasized how the work is the first one to apply the method in the smart building literature, specifically in the context of energy storage and cash flows from reserve power markets (e.g. on lines 73-78).

Reviewer 6 Report

Viability of a virtual power plant in a smart building system – A techno-economic perspective is presented. The methodologies and results in this study may be of interest to researchers in cognate fields and the outcomes of this work may benefit future studies. However, there are some major issues that must be addressed.

This paper reasonably well written. However, some of the sentence structuring is clunky and there are numerous grammatical errors. The writing in this paper can be improved and the English polished.

The abstract is lacking information on the results of the study. The reader should be presented with some key quantitative findings.

Even though several papers are referenced and briefly summarised in the in section 1, it is difficult to get a feel for where this study lies in the context of previously published work from the lit review in this section. A more comprehensive and holistic lit review in regards to the state of the art and the application of cognate techniques would give a feel for where this research lies in regards to past studies. The author should review popular methods of solving similar problems in cognate applications. The author must explicitly delineate his/her method from the existing ones, and must explicitly state the benefits of this method to clearly define the original contribution of this paper.

The novel contribution of the paper must be clearly defined. As there have been so many paper published in this area in the last 10 years, it is very important to clearly state the original contribution to the field.

In order for this paper to make a contribution to the field, the experimental data should be openly available. Please provide a link to the virtual power plant model/program.

A detailed diagram/plan of the building with a detailed breakdown of its properties is required.

Is this really a virtual power plant? It is more of an SRI assessment. I think the title maybe misleading, I was expecting electrical models.

The data collection method is not adequately described.The data section is lacking detail. It gives very little information on the scale, resolution or granularity of the data used for this analysis.

A detailed stats analysis of the recorded data is required. 

All parameter choices and assumptions must be justified. All parameters and assumptions used should be displayed in a table in the appendix for reproducibility purposes.

Data visualisations are really needed in this paper. The limited number of tables do not provide enough information. It is not intuitive, the reader does not really digest the results presented.

Where is the critical analysis? There should be a proper discussion.

Cognate works should be compared and discussed in the discussion section.

What are the limitations of this work?

What general outcomes can be applied to other similar studies?

In the conclusion, avoid giving a general summary of the salient points. Make sure to give actual conclusions.

At present, I do not see the novel contribution this paper makes to the field. The author must address this in the revised manuscript.

Major corrections are required before this paper can be considered for publication.

Author Response

Thank you for your valuable comments.

General:

  • We have done major corrections and aimed to clarify the content of the paper.
  • The content of the paper has been refined towards economic analysis.
  • We also changed the title in order to make the content and nature of the paper more explicit.
  • The novelty of this paper has been highlighted, e.g. on lines 80, 90, 727 868, 998, and 1042
  • English has been polished.

Abstract:

  • The key findings of the paper have now been introduced in lines 18-19.

Section 1: Introduction

  • The existing literature has been reviewed, and related research from the field has been introduced starting from line 58.
    • The popular methods to solve similar problems including payback period and return on investment have been introduced (on lines 63, 65, and 66).

  • To the best of our knowledge the discounted cash flow analysis has not been applied in similar (smart energy building) studies before. However, the method has been successfully applied in on-site energy investment studies, which have been stated in lines 66-69.

  • Additionally, the benefits of the general method (DCF) have now been described better. It has now been emphasized how the work is the first one to apply the method in the smart building literature, specifically in the context of energy storage and cash flows from reserve power markets (e.g. on lines 76-78).

  • The novelty has now been clearly introduced  (e.g. on lines 76-78 and 558-563)

Section 2: Research design

  • We have refined the research design and aimed to describe the data collection explicitly. The focus of this paper is on the economic analysis of a real-life smart (energy) system investment in a building.
  • The originality of the paper is now better described. The present paper is not an experimental paper, but an empirical paper with real-life smart building system with measured cash flows (smart investments, energy savings, and reserve market income).
  • The paper does not study a virtual power plant configuration but verified incomes from smart energy systems for the property owners. Due to confidential reasons, the virtual power plant program from the commercial service provider is not available, but the monetary data from the smart energy system of the property (previously called as a virtual power plant) was retrieved from the building system and verified based on the service agreement.

  • The economic data utilized in the paper has been introduced.

  • The aim and originality have been now restated so that it does not give a misleading impression of being a virtual power plant study, but novel economic investment analysis of a real-life smart building with identified new cash flows.

  • We changed the title and left the techno-perspective out of the scope to make the content more explicit.

  • In this paper, we applied the SRI assessment to validate the (energy) smartness of the building system.
  • The choices made in the economic analysis have been justified and references provided where needed.

Section 3: Results

  • The content of the paper has been refined, and a clearer distinction between SRI assessment, investment calculations, and interviews has been made.

Section 4: Discussion

  • Critical analysis has been provided starting from line 520.

  • To best of our knowledge, this is the first study to apply a DCF analysis on a smart energy system investment. We have now aimed to provide a proper discussion of the limitations and benefits of the applied method.

  • Limitations have been discussed, starting from line 540.

Section 5: Conclusions

  • In the present study, the investment profitability was evaluated through the impact of the savings in operating expenses and additional income (generated by the investment) on the property value. We suggest this approach to be used in other similar studies as stated in lines 572-575.

  • We have refined the conclusion and provided quantitative results to support the qualitative ones.   

Round 2

Reviewer 3 Report

Thank you for incorporating my comments and improving the paper.
It is now clearer what the aim is.

Please check equation (6) again.
One of the summands in the numerator just shows i=20 as the lower bound of the summation, but no upper bound. It seems that i=20 is the only value such that the summation is not necessary.
Moreover, there is a summation from i=1 to 30 before the fraction.
Is this correct? Shouldn't this be in the denominator only?

Author Response

Thank you for your valuable comments and recommendations to improve the content of the paper. We have now aimed to make the final improvements to the present paper. 

  • Thank you for the notice. The equation (6) has now been refined. 

  • The function is correct. In this type of DCF analysis (in which the aim is to evaluate the impact of smartness on property value through savings in operating expenses and additional income) the analysis is two-folded.
    1. First, we include the foreseen inflation rate on cash flows (30 years analysis).
    2. Secondly, the value generated through the location is incorporated into the analysis by using the sum of the property’s yield (4.5%) and the inflation rate (1.5%). This sum was applied as a discount rate when capitalizing the net savings and new income (generated by the smart investment) into the value of the property. 

Besides the provided comments, we have now further analyzed cognate studies from the field in the introduction. Additionally, we have elaborated more in detailed the benefits of the chosen method (DCF) compared to the regularly applied investment metrics (on lines 58-88).

Reviewer 4 Report

The article has improved significantly since the last revision, however some of the comments made in the previous revision have not yet been addressed and therefore the article cannot be accepted in the present form.

- It is necessary to include a complete analysis of the previous work done by the scientific community and case studies similar to the current one, even if with other methods.

- In addition, I believe that it is necessary to include an analysis of new technologies and the implications they would have on a smart building. This type of technologies are basic for this type of buildings if their impact in the implementation is not analyzed, the study is totally incomplete.

Author Response

Thank you. We have now aimed to address the final comments.

  1. The analysis of cognate works has now been further elaborated on lines 58-88. Additionally, the benefits of the chosen method (DCF) have been described more in detail in the introduction.
  2. Thank you for the recommendation. However, in the first revision round the aim and originality of the study was refined, so that it does not give a misleading impression of being a virtual power plant (i.e. technological) study. Thereafter, we would like to highlight that the present paper is not experimental, but an empirical paper of a real-life smart building system with measured cash flows (smart investments, energy savings, and reserve market income). The paper does not study smart technologies and their implications on the building performance but verifies the incomes (enabled by the smart energy system) for a property owner. Thereafter, to maintain the clarity of the paper, we do not see the need for an analysis of the new technologies. 

Reviewer 6 Report

The authors have not adequately addressed the first round comments. Each comment should be directly addressed. 15 comments were provided in the last round, howver they were not all replied to. THe reviewers most procive a comrephensive report explicititly stating how each comment was dealt with. 

Author Response

We have now aimed to clarify the report. Please see below the answers, which have now been listed in numbers based on the comments given in the first round.

  1. Comment: The abstract is lacking information on the results of the study. The reader should be presented with some key quantitative findings.
    Answer: The key findings of the paper have now been introduced in the abstract on lines 18-19.

  2.  
    1. Comment: A more comprehensive and holistic lit review in regards to the state of the art and the application of cognate techniques would give a feel for where this research lies in regards to past studies.
      Answer: The existing literature has been reviewed, and the related research from the field has been introduced. Further analysis of the existing work has been provided on lines 58-88.
    2. Comment: The author should review popular methods of solving similar problems in cognate applications.
      Answer: The popular methods to solve similar problems have been reviewed, including payback period and return on investment on lines 62-65.

    3. Comment: The author must explicitly delineate his/her method from the existing ones, and must explicitly state the benefits of this method to clearly define the original contribution of this paper.
      Answer: To the best of our knowledge the discounted cash flow analysis has not been applied in similar (smart energy building) studies before. However, the method has been successfully applied in on-site energy investment studies, which have been stated on lines 83-85.
      The insufficiency of the regularly applied methods has been stated on lines 65-67 and 79-83.
      The benefits of the proposed method (DCF) have now been described better on lines 68-77. Also, it has now been emphasized how the work is the first one to apply the method in the smart building literature, specifically in the context of energy storage and cash flows from reserve power markets.
  3. Comment: The novel contribution of the paper must be clearly defined. As there has been so many paper published in this area in the last 10 years, it is very important to clearly state the original contribution to the field.
    Answer: The novelty has been introduced  (e.g. on lines 78-81 and 87-88).

    1. Comment: In order for this paper to make a contribution to the field, the experimental data should be openly available. 
      Answer: The originality of the paper is now better described in the introduction. The present paper is not experimental, but an empirical paper with real-life smart building system with measured cash flows (smart investments, energy savings, and reserve market income). The paper does not study a virtual power plant configuration but verified incomes from smart energy systems for the property owner.

    2. Comment: Please provide a link to the virtual power plant model/program.
      Answer: Due to confidential reasons, the virtual power plant program from the commercial service provider is not available, but the monetary data from the smart energy system of the property (previously called as a virtual power plant) was retrieved from the building system and verified based on the service agreement. The economic data utilized in the paper has been introduced in Table 3.

  4. Comment: A detailed diagram/plan of the building with a detailed breakdown of its properties is required.
    Answer: We changed the title and left the techno-perspective out of the scope to make the content more explicit. The content has been refined (to be align with the title), so we do not see a need for a detailed technological breakdown plan of the building.

  5. Comment: Is this really a virtual power plant? It is more of an SRI assessment. I think the title may be misleading, I was expecting electrical models.
    Answer: The aim and originality have been now restated so that it does not give a misleading impression of being a virtual power plant study, but novel economic investment analysis of a real-life smart building with identified new cash flows. In this paper, we applied the SRI assessment to validate the (energy) smartness of the building system.

  6. Comment: The data collection method is not adequately described. The data section is lacking in detail. It gives very little information on the scale, resolution, or granularity of the data used for this analysis.
    Answer: The aim of the paper was not to evaluate the technical capability of the smart energy system but rather assess the investment profitability in financial terms. Thus, data collection is explicitly described in section 2.
    • First, the performed SRI assessment is described in section 2.2. The SRI assessment was applied to validate the smartness of the system as described in lines 137-138.
    • Second, the investor interviews were applied to figure out the decision-making process of such a smart investment, as well as to validate the economic analysis performed by the authors.
    • Third, the economic analysis of the investment was performed. In the analysis, we applied investment data that is publicly available. We have introduced the applied data in Table 3. New income and savings (which were generated by the investment) were based on 10 months of actually running the system. This has been stated on lines 252-253. However, due to confidentiality reasons, the data gathered from the virtual power plant model is not publicly available.

  7. Comment: A detailed stats analysis of the recorded data is required. 
    Answer: Stats analysis was not performed in the present study. The data was retrieved from the service supplier, who was in charge of the technical analysis of the smart energy system.

  8. Comment: All parameter choices and assumptions must be justified. All parameters and assumptions used should be displayed in a table in the appendix for reproducibility purposes.
    Answer: All parameter choices as well as assumptions have been justified. The parameters and assumptions applied to make the economic analysis have been stated in section 2.3.1 and summarized in Table 3.
    To increase the transparency of the conducted economic analysis, in the first revision round the equation applied to calculate the expected property value increase was added with inputted economic variables (equation 6 on lines 302-305). Thus, we do not see a need for an Appendix in the scope of the present paper.

  9. Comment: Data visualizations are really needed in this paper. The limited number of tables do not provide enough information. It is not intuitive, the reader does not really digest the results presented.
    Answer: As the aim of the paper was refined, we do perceive the number of tables as adequate in the context of the paper scope. 

  10. Comment: Where is the critical analysis? There should be a proper discussion.
    Answer: Critical analysis has been provided in the discussion section e.g. starting from line 543.

  11. Comment: Cognate works should be compared and discussed in the discussion section.
    Answer: To best of our knowledge, this is the first study to apply a DCF analysis on a smart energy system investment. In the discussion-section, we have aimed to provide a proper discussion of the limitations and benefits of the applied method. Cognate works have been discussed e.g. on lines 493-502. 

  12. Comment: What are the limitations of this work?
    Answer: Limitations have been discussed, starting from line 532.

  13. Comment: What general outcomes can be applied to other similar studies?
    Answer: In the present study, the investment profitability was evaluated through the impact of savings on operating expenses and additional income (generated by the investment) on property value. We suggest this approach to be used in other similar studies as stated on lines 586-589.

  14. Comment: In the conclusion, avoid giving a general summary of the salient points. Make sure to give actual conclusions.
    Answer: We have refined the conclusion and provided quantitative results to support the qualitative ones on lines 577-579.